# Impact of Polyethylene Glycol Functionalization of Graphene Oxide on Anticoagulation and Haemolytic Properties of Human Blood

**DOI:** 10.3390/ma14174853

**Published:** 2021-08-26

**Authors:** Trayana Kamenska, Miroslav Abrashev, Milena Georgieva, Natalia Krasteva

**Affiliations:** 1Institute of Biophysics and Biomedical Engineering, Bulgarian Academy of Sciences, “Acad. Georgi Bonchev” Street Bl. 21, 1113 Sofia, Bulgaria; trayanakamenska@abv.bg; 2Faculty of Physics, Sofia University “St. Kliment Ohridski”, 5 James Bourchier Blvd., 1164 Sofia, Bulgaria; mvabr@phys.uni-sofia.bg; 3Institute of Molecular Biology “Acad. R. Tsanev”, Bulgarian Academy of Sciences, “Acad. Georgi Bonchev” Street Bl. 21, 1113 Sofia, Bulgaria; milenageorgy@gmail.com

**Keywords:** nano-graphene oxide (nGO), nGO-PEG, haemocompatibility, activated partial thromboplastin (APTT), prothrombin time (PT), nanoparticles (NPs)

## Abstract

Graphene oxide (GO) is one of the most explored nanomaterials in recent years. It has numerous biomedical applications as a nanomaterial including drug and gene delivery, contrast imaging, cancer treatment, etc. Since most of these applications need intravenous administration of graphene oxide and derivatives, the evaluation of their haemocompatibility is an essential preliminary step for any of the developed GO applications. Plentiful data show that functionalization of graphene oxide nanoparticles with polyethylene glycol (PEG) increases biocompatibility, thus allowing PEGylated GO to elicit less dramatic blood cell responses than their pristine counterparts. Therefore, in this work, we PEGylated graphene oxide nanoparticles and evaluated the effects of their PEGylation on the structure and function of human blood components, especially on the morphology and the haemolytic potential of red blood cells (RBCs). Further, we studied the effect of PEGylation on some blood coagulation factors, including plasma fibrinogen as well as on the activated partial thromboplastin (aPTT), prothrombin time (PT) and platelet aggregation. Our findings provide important information on the mechanisms through which PEGylation increases GO compatibility with human blood cells. These data are crucial for the molecular design and biomedical applications of PEGylated graphene oxide nanomaterials in the future.

## 1. Introduction

Nanoparticle-based strategies have great promise to revolutionize contemporary medicine. Various nanoparticle-based strategies are used in the prevention, diagnostics and therapy of many diseases such as cardiovascular, neurodegenerative, infectious and autoimmune, cancer, hepatitis, etc. [1,2,3,4]. Even though many different nanoparticles are already approved by the U.S. Food and Drug Administration (FDA) and are on the market, many are in clinical trials [5,6]. Therefore, the design and improvement of certain strategies for the application of nanoparticles in biomedical practice are continuously improving [7]. Consequently, intensive research has been conducted to enhance water dispersibility, biocompatibility, renal clearance and to diminish the toxicity of nanoparticles (NPs) [8]. Surface modification is an easily achievable, simple and efficient approach to endow new properties and functions to the surface of nanomaterials whilst retaining their original bulk properties [9,10,11,12,13,14,15].

Graphene oxide (GO) is one of the most widely explored and exploited nanomaterials of recent years [16,17]. It finds applications in numerous biomedical uses including drug and gene delivery, contrast imaging, biosensors, and photothermal therapy [18,19]. It is a 2D carbon nanomaterial with a large surface-area-to-volume ratio, high stability, good biocompatibility, and an easy surface modification capacity [20,21,22]. The presence of graphene *sp*^2^ domains in the structure of GO defines its noticeable broadband fluorescence ability, while the hydroxyl and epoxy groups on the basal plane of the sheets and carboxyl groups at the edges allow its easy surface modification with different functional groups and organic molecules including drugs which covalently conjugate with GO [23]. Non-covalent functionalization of GO surfaces (by π–π interactions or electrostatic, van der Waals interactions) allows successful attachment of DNA, peptides, polymers and other cell-targeting antibodies and molecules [24].

Generally, all NP applications in biomedical practice require evaluation of two very important features of the NPs: their cellular toxicity and uptake [16]. The haemocompatibility of NPs is one of the most critical characteristics for safe applications because most of them need intravenous administration [15,25]. Conversely, the small size of NPs allows translocation into the circulatory system through dermal contact, inhalation or ingestion systemic administration [26,27]. Once entering the blood circulation, NPs come into contact with blood cells and plasma proteins and may induce activation or destruction of blood components, eventually affecting the whole organism [26,28]. Therefore, the haemocompatibility of the blood-contacting nanomaterials is exceedingly important, highlighting the importance of their careful analysis. Additionally, knowledge about nanomaterials’ haemocompatibility will accelerate the development and clinical application of biomaterials with optimal coagulation and haemostatic properties. A good example is the newly improved wound dressing based on nanocellulose and alginate fibres with wound healing promoting properties [29,30].

According to ISO-10993-4, the evaluation of blood compatibility of all nanomaterials includes assessment of haemolysis, platelet activation, coagulation, fibrinolysis, fibrin formation, thrombosis and complement activation [31]. Haemolysis is the destruction of erythrocytes (red blood cells, RBCs) leading to a release of their contents (e.g., haemoglobin) into the surrounding tissues [26,32,33]. Nanomaterials that enter the blood come in contact with red blood cells (RBCs) and might affect the membrane integrity of RBCs, which may lead to pathological conditions such as anaemia, hypertension and renal failure [26]. Platelets, also known as thrombocytes, play a key role in haemostasis [26]. In blood circulation, platelets can quickly adhere and aggregate at sites of vascular injury, forming a platelet plug (i.e., the first wave of haemostasis). A homeostatic imbalance in platelet function (primary haemostasis) or the coagulation system (secondary haemostasis) can lead to thrombotic or haemorrhagic disorders [34]. Nanomaterials can interfere with some of these reactions, as has been recently demonstrated [35]. Additionally, NP-induced coagulopathy has become a serious concern, as several studies reported an increased risk of cardiovascular disease due to NP-induced thrombotic complications [25]. The complement system can also be activated by contact with nanomaterials, mainly via the classical pathway or the alternative pathway, which leads to the generation of the anaphylatoxins C3a and C5a, and promote leukocyte activation and pro-inflammatory response [29].

Pristine GO nanoparticles have a great risk of intravenous toxicity because of the small size of the nanoparticles, which permits the binding of more proteins per mass than larger particles [36,37]. These interactions can lead to extensive haemolysis, platelet aggregation, complement activation, inflammation and fast clearance of the particles from the circulation [38,39,40,41]. Additionally, GO NPs easily agglomerate under physiological conditions because of high specific surface area. Agglomeration greatly decreases blood circulation and results in a decrease in the loading capability of bioactive agents. To address these issues, stable and dispersible GO nanoparticles have been developed by functionalization with various materials, such as proteins, naturally abundant polysaccharides and polymers [41]. Poly(ethylene) glycol (PEG) is considered as one of the most promising surface stabilizing materials for GO because of its multifunctionality, biocompatibility, nontoxicity, high natural abundance and renewing ability [42,43]. PEG creates a steric shield around the particles, preventing interactions among the particles and the blood components [44].

In the current work, we functionalized nGO NPs with PEG to obtain physiologically stable nanoparticles with good water dispersibility and low blood toxicity. To understand the in vitro safety and efficacy of the nGO-PEG nanoparticles, their haemocompatibility was investigated—by assays that aim to detect their impact on blood clotting, investigated by measurement of the activated partial thromboplastin (APTT), prothrombin time (PT) and fibrinogen. In addition, the platelet aggregation along with the lysis and morphology of human red blood cells (RBCs) were further examined. The obtained results highlight the increased haemocompatibility of the PEGylated GO NPs, thus opening new opportunities for the safe blood administration of PEGylated GO in various medicinal practices.

## 2. Materials and Methods

### 2.1. Synthesis of Poly (Ethylene Glycol) Graphene Oxide (nGO-PEG) and Physicochemical Characterization

Functionalization of GO with PEG was performed as previously described [45,46]. Briefly, pristine GOs (Graphenea, San Sebastian, Spain) with a concentration of 1 mg/mL were sonicated for 2 h at 500 W (VCX 500, Sonics and Materials, Inc., Newtown, CT, USA) to obtain nanosized graphene oxide (nGO) NPs. mPEG-NH_2_ (Abbexa Ltd., Cambridge, UK) was added. The obtained mixture was sonicated for 5 min and left overnight at 70 °C in a water bath. The resulting nGO-PEG suspension was centrifuged at 13,000× *g* for 20 min to remove any unstable aggregates and stored at 4 °C. For haemocompatibility studies, both nGO and nGO-PEG NPs were sonicated for 1h in a water bath (UM-2, Unitra, Lipsk, Poland) just before experiments.

Both as-prepared nGO and nGO-PEG NPs were characterized by Specord 210 Plus spectrophotometer (Edition 2010, Analytik Jena AG, Jena, Germany), Raman spectrometer (LabRAM HR, HORIBA Jobin Yvon, Villeneuve d’Ascq, France), TEM (JEM-2100, Tokyo, Japan) and DLS Zetasizer (Malvern Instrument, Ltd., Malvern, UK) to measure UV–VIS-absorption spectra, Raman spectra nanoparticle morphology, zeta potential and size, and particle distribution, respectively.

### 2.2. Blood Samples Collection

Fresh blood samples were obtained from healthy volunteers who did not take any medicines known to affect platelet and erythrocyte functions at least two weeks before studies. The blood samples were obtained, for scientific research only, after written informed consensus, as required by the Bulgarian “Bioethics Committee.” All blood preparation methods and sample manipulation were performed by trained personnel and following the relevant guidelines and regulations approved by the “Bioethics Committee” at the Institute of Neurobiology, BAS. Blood samples were collected in blood collection tubes containing heparin or sodium citrate as anticoagulants and were used within 24 h.

### 2.3. Haemolytic Characterization

Haemolytic experiments and morphological observations were performed on red blood cells (RBCs), isolated from the whole blood, containing heparin or sodium citrate as anticoagulation agents. Briefly, the whole blood was centrifuged at 1000× *g* rpm (at Hettich EBA-20 centrifuge, Tuttlingen, Germany) for 5 min and the upper layer of plasma was carefully removed. The remaining RBC pellet was washed three times with PBS, re-dispersed with PBS to initial volume and was further used for the experiment.

#### 2.3.1. Haemolysis Assay

To evaluate the haemolytic activity of nGO and nGO-PEG NPs, NPs with different concentrations were added to RBCs and isolated from whole blood containing heparin as an anticoagulant to a final volume of 0.3 μL. For positive and negative controls, cells exposed to a 10% Triton X-100 solution (+RBCs) and PBS (+RBCs), respectively, were used. The samples were placed in an incubator at 37 °C for 3 h and were shaken every 30 min. After incubation, samples were centrifuged for 5 min at 1000× *g* rpm (Hettich EBA-20 centrifuge) and haemoglobin absorbance in the supernatant of the samples was measured at an automated blood coagulation analyzer (ACL 7000, Instrumentation Laboratory, Bedford, MA, USA).To quantify the percentage of haemolysis, the measured haemoglobin concentration was divided by the haemoglobin concentration of the diluted blood solution as described by the following equation:% Haemolysis = Haemoglobin conc. of sample/Haemoglobin conc. of diluted blood × 100

#### 2.3.2. Morphological Observation of Red Blood Cells (RBCs)

Red blood cell morphological observations were conducted on isolated RBCs from fresh citrated human whole blood. After isolation of the RBCs, the suspension was incubated with nGO and nGO-PEG NPs with different concentrations for two hours at 37 °C. At different time points (0, 30, 60 and 120 min), the RBCs were deposited on glass slides, air-dried and stained with Giemsa stain solution for 30 min, washed with deionized water (D.I.), air-dried again and observed under a phase-contrast microscope (Olympus CH-2 Refurbished Microscope, Tokyo, Japan).

### 2.4. Plasma Coagulation Studies

#### 2.4.1. Measurement of Plasma Coagulation Time and Fibrinogen Concentration

Platelet-poor plasma (PPP) was prepared through centrifugation of fresh citrated human whole blood at 3000 rpm for 15 min and then the resulting supernatant was collected. GO NPs with different concentrations were added to PPP to a final volume of 300 μL. All samples were incubated at 37 °C for 30 min and then analyzed by an automated blood coagulation analyzer (ACL 7000, Instrumentation Laboratory, Bedford, MA, USA) for prothrombin time (PT), activated partial thromboplastin time (aPTT) and fibrinogen concentration (FNG).

#### 2.4.2. Platelet Aggregation Assay

Platelet-rich plasma (PRP) was obtained through centrifugation of fresh human whole blood collected in a BD Vacutainer ACD-A (REF 366645) at 500× *g* for 5 min, and the upper layer of plasma was carefully transferred to a tube containing sodium citrate. After another centrifugation at 700× *g* for 7 min, the resting pellet was resuspended with 1.5 mL PBS. NPs with a concentration of 100 μg/mL were added to PRP to a final volume of 300 μL. PBS was used as a negative control and adenosine diphosphate (ADP) as a positive control. ADP (50 μM/mL) was also added to half of the samples as an agonist to promote platelet aggregation. The samples were shaken for 15 min at 37 °C and then the phase-contrast photos were taken under Olympus CH-2 refurbished microscope.

### 2.5. Statistical Analysis

Experiments were in triplets and data are presented as MEAN values ± STDV (mean standard error). The statistical differences were analyzed by the Student’s *t*-test. A statistically significant difference was defined for *p*-values < 0.05.

## 3. Results

### 3.1. Functionalization and Characterization of nGO and nGO-PEG

PEGylation of GO was performed following the one-step protocol of Chen et al. for simultaneous reduction and PEGylation of GO [47]. A detailed description of the procedure with modifications’ details can be found in [45,46]. We have decreased the reduction temperature from 90 °C to 70 °C to avoid abundant reduction of oxygen functional groups at the surface of GO, which is needed for further modification of GO with drugs and other biomolecules. A two-hour ultrasonication of GO flakes was performed before modification with PEG to decrease the particle size, thus improving particles’ penetration into the blood cells. Biomedical applications of nanoparticles require a good understanding of their physicochemical properties [48]. We have used different techniques to characterize the nGO and nGO-PEG nanoparticles’ structure and morphology, zeta potential, particle size distribution and hydrodynamic diameter to confirm the successful functionalization of nGO with PEG (Figure 1). Transmission electron microscopy (TEM) was first performed, and the obtained micrographs proved the successful functionalization of nGO with PEG (Figure 1A). TEM nGO and nGO-PEG images displayed an irregular form with the typical transparent, sheet-like morphology of graphene. However, PEGylation altered GO’s morphology, by folding GO’s sheets and by the formation of wrinkles probably due to the spatial interactions between the PEG molecules grafted onto the edges of the GO layers. Other previous studies have also been reported on a more layered and folded morphology of GO after functionalization [49,50]. Raman spectroscopy was further used for studying graphene structural changes upon functionalization [51]. The Raman spectra of nGO and nGO-PEG are shown in Figure 1B. Raman spectra proved the increase in nitrogen content in nGO-PEG samples. In our samples D, D’ and D+G bands appeared in addition to G bands (1580 cm^−1^). The G band was indicative of *sp*^2^-hybridized carbon atoms and assured a preserved graphitic lattice after functionalization of nGO with PEG. The D and D+G peaks referred to *sp*^3^-hybridized carbon systems and proved the presence of defects characteristic of the graphene oxide-based materials [52]. The absence of the 2D band, which appeared as the most intense feature in perfect single-layer graphene, or an extra-wide band also indicated that GO was dominated by the high degree disordered structure. The nGO-PEG spectrum revealed a slight blue-shift of the 2D band and red-shift of the D-band compared to nGO, which together with the appearance of the D’ band, proved the increase in nitrogen incorporation in the sample. The intensity ratio ID/IG was commonly reported to assess the *sp*^2^ domain extent in GO materials [53]. The starting material GO scored a ratio of 0.947, while in nGO-PEG it was 0.927, suggesting recovery of aromatic structures during PEGylation by repairing defects. The hydrodynamic diameters and size distribution of nGO and nGO-PEG in D.I. water were determined by dynamic light scattering (DLS) and are shown in Figure 1C. Although the DLS characterization did not reveal the exact size of these GOs particles in aqueous solution because of the anisotropic morphology of GOs, the DLS results showed that the average hydrodynamic size of nGO-PEG increased after PEGylation from 252.7 nm for nGO to 324.6 nm of nGO-PEG. This was another indicator for the successful modification of nGO with PEG. The surface charge of nanoparticles plays an important role in blood component–nanoparticle interactions, because blood cell membranes and blood proteins are charged. Measurements of the surface charge of nGO and nGO-PEG particles (Figure 1D) showed that the zeta (ζ) potential of both types of NP was negative: −32.9 ± 7.24 mV for nGO and −21.6 ± 5.63 mV for nGO-PEG. The lower ζ potential of PEGylated nGO was consistent with the decreased oxygen amount as a result of reduction and functionalization of GO with PEG and confirmed successful coating of PEG onto nGO surface.

Haemocompatibility of nGO and nGO-PEG nanoparticles was analyzed on anticoagulated human whole blood, derived from healthy volunteers by evaluation of the effect of both graphene systems on haemolysis, erythrocyte cell morphology, platelet aggregation, coagulation time and coagulation factors.

### 3.2. Haemocompatibility of nGO and nGO-PEG NPs

#### 3.2.1. Haemolytic Activity of nGO and nGO-PEG NPs

Haemolysis refers to the release of haemoglobin (Hb) from RBCs and indicates a disturbance in RBC membrane integrity which may lead to different adverse effects on human health (e.g. anaemia, hypertension, renal toxicity) [54]. All nanomaterials that enter the blood come in contact with red blood cells and might affect the membrane integrity of RBCs by mechanical damage or generation of reactive oxygen species (ROS). Therefore, the haemolytic potential of all nanoparticles contacting with the blood needs to be evaluated [26]. The haemolysis test in the current study was performed by spectrophotometric measurement of the released haemoglobin in plasma that was derived from blood, treated with increasing concentrations of NPs for 3 h and subsequent centrifugation. Non-treated blood, diluted with PBS, was used as a negative control (marked as PBS), and as a positive control, Triton X-100-treated whole blood was applied (marked as Triton X-100). In the negative (PBS) control Hb release was not registered, while in the positive control (Triton X-100) haemolysis was almost 100%, which confirmed the validity of the assay. Figure 2A shows the haemolysis percentage of RBCs exposed to different concentrations of nGO and nGO-PEG NPs as a function of the used NP concentrations. The haemolysis rate was very low in blood samples treated with 10 μg/mL GO NPs, indicating no detectable disturbance of the RBC membranes, and remained low for concentrations up to 50 μg/mL. With increasing NP concentrations over 100 μg/mL the haemolysis percentage of RBCs increased, and at a concentration of 200 μg/mL, the haemolysis was 10.7% and 8% for nGO and nGO-PEG, respectively, suggesting that the higher concentrations of GO NPs resulted in rupture of the RBC membrane. The haemolysis percentage of RBCs increased when the concentration exceeded 100 g/mL. This indicated that excessive concentration increased the rate of haemolysis. According to ASTM E2524-08 Standard-Standard Test Method for Analysis of Haemolytic Properties of Nanoparticles (ASTM International, West Conshohocken, PA, USA, 2000), haemolysis higher than 5% is considered significant RBC lysis [33]. Therefore, the higher concentrations of nGO and nGO-PEG nanoparticles induced haemolysis in RBCs and could be considered haemotoxic. The lower percentage of Hb release in nGO-PEG blood treated samples suggested that PEGylation diminished the haemolytic activity of nGO. The presented optical images of the blood samples in Figure 2B,D confirmed the spectrophotometric measurements. The dark red colour of Triton-treated blood samples was a result of approximately 100% leakage of Hb into the supernatant; the yellow colour of PBS-treated samples indicated an absence of haemolysis and released Hb. The different colours of NP-treated RBCs from yellow to red correspond to different degrees of haemolysis and Hb release in the blood samples.

Our results are in accordance with a work previously published by Feng and co-workers indicating that graphene oxide and graphene sheets (GS) exhibited dose-dependent haemolysis [55]. However, the authors reported on a very high haemolytic activity of GO particles: 2.45% haemolysis in blood samples treated with 0.001 mg/mL GO; >5% haemolysis in the presence of 0.01 mg/mL of GO and up to 96% haemolysis in 0.5 mg/mL GO-treated blood samples. In contrast, Sasidharan et al. showed that pristine (p-G) and functionalized graphene (f-G) exhibit a very high haemocompatibility, causing less than 0.2% haemolysis in RBC in concentrations up to 75 μg/mL, probably due to much lower surface area and hydrophobic surface [56]. The differences between our results and those of the other authors are probably because of the difference in physicochemical properties of GO NPs as a result of different preparation methods. The interaction of GO NPs with RBCs’ surface should be driven by hydrophobic interaction between the hydrophobic part (graphene skeleton) of GO and phospholipid molecules on RBCs’ membrane, which caused aggregation, morphological change, and membrane integrity damage of RBCs. In addition, other nonspecific interaction forces such as hydrogen bonding between carboxyl groups in the glycocalyx of RBCs and hydroxyl and carboxyl groups of the GO could also take place and contribute to the interaction. Further, we observed a diminished haemolytic activity in PEGylated nGO, revealing that PEG serves as a protective layer, masking the electrostatic interactions between RBCs and oxygen groups on the nGO surface, and decreasing the cell-contactable surface area [57]. The differences in haemolytic properties of nGO and nGO-PEG may be due to the differences in their size and particle size distribution. Particle size, together with chemical composition, is one of the most important characteristics that influences interactions with cells [36,37]. The observed lower haemolytic activity of nGO-PEG as compared to that of nGO is due to the larger particle size and distribution resulting in a small number of NPs contacting with cells and destroying erythrocyte membrane integrity. On the other hand, in the case of nGO NPs, a great number of NPs have a very small size. These NPs can very easily damage the cell membrane because of the sharper edges of pristine nGO NPs, which results in a higher degree of haemolysis.

#### 3.2.2. Effects of nGO and nGO-PEG NPs on Erythrocyte Morphology

In addition to haemolysis, exposure of RBCs to nanoparticles may induce alteration in cell morphology, resulting in swollen cells, stomatocytes, echinocytes and haemagglutination [58,59]. These altered morphologies are frequently indicative of various medical conditions [60]. To understand better the effect of PEGylation on RBCs, we have studied the morphologies of RBCs after being treated by different concentrations (0, 10, 20, 50, 100 and 200 μg/mL) of nGO and nGO-PEG NPs for different periods (0, 30, 60 and 120 min). The results are shown in Figure 3. Compared with the normal RBCs with a biconcave shape and a smooth membrane in the control samples, the exposure to nGO and nGO-PEG NPs led to spherocytic/flat shaped erythrocytes, cells with an irregular shape (ovalocytes, teardrop cells, sickle cells), cells with numerous surface spikes (echinocytes), and some ghost cells, which are the result of cell lysis and release of haemoglobin from RBCs. Substantial differences between both types of GO NPs, however, were not found, and nor were detectable concentration- and time-dependent morphological alterations in RBCs. With the time of incubation with GO NP RBCs’ morphology becomes spikier and agglutinated as in the pristine nGO-treated group these morphological changes were observed after 60 min incubation, while in nGO-PEG-treated blood samples the alteration in RBCs’ morphology was delayed and occurred at 120 min. This confirmed the positive effect of PEGylation on the haemolytic properties of nGO.

Morphologically altered RBCs, however, have been observed also in the control groups at different time points, suggesting an influence of other factors different from NP exposure which contribute to variation in erythrocyte morphology, such as prolonged incubation time. The loss of normal biconcave shape and morphological variations in RBCs suggest a haemolytic activity and potential toxic effect of nGO and nGO-PEG which needs further investigations for an in-depth understanding of the mechanism. It should be kept in mind that the presented images are only from one healthy donor; thus, the effect of GO NPs could not be generalized, and a correlation with haemolytic activity is not fully correct.

Indeed, the literature data on the effect of GO NPs on erythrocytes’ morphology are very controversial. Most of the studies showed that the GO influences RBCs’ morphology and membrane integrity in a concentration-dependent manner, as the higher concentration of GOs leads to the formation of echinocytes, whereas lower concentration leads to spherocytic shaped erythrocyte [55,56,57,58,59]. The other studies, however, did not show any harmful effect of NPs on human RBC structure [60].

### 3.3. Anticoagulation Properties of nGO and nGO-PEG NPs

Coagulation, or blood clotting, is one of the most important functions of the blood. It is a complex physiological process by which the blood forms a blood clot, thus preventing blood loss and bleeding. The coagulation system includes the endothelial lining of the blood vessels and cellular (platelets) and soluble (plasma proteins) components from the blood. Normal coagulation in the body is maintained by the balance between the pro-and anti-coagulation pathways responsible for the formation and inhibition of the clots. The imbalance of the coagulation system may lead to thrombosis or bleeding, which may cause severe, even lethal damage to the living organism. Biomaterials contacting with blood often act as agonists or activators of the coagulation system by interacting with the components of the coagulation system (e.g., clotting factors, fibrinogen and platelets) and may adversely affect the clotting process [61]. Therefore, plasma coagulation studies are very important to evaluate the haemocompatibility of the biomaterials. Here, we have assessed the effect of nGO and nGO-PEG NPs’ coagulation time for intrinsic and extrinsic pathways, platelet (thrombocytes) aggregation and fibrinogen concentration, after exposure to platelet-poor plasma (PPP) and platelet-rich plasma (PRP), respectively, to the increasing concentrations of NPs for 30 min. Results are shown in Figure 4.

#### 3.3.1. Effect of nGO and nGO-PEG NPs on Patients’ Blood Coagulation Time

Plasmatic coagulation is classified into intrinsic and extrinsic pathways. In normal haemostasis, the primary task of the extrinsic pathway is initiating haemostasis. The intrinsic pathway acts as ‘executor’ of blood clotting but can be activated separately by various triggers, typically natural and artificial surfaces (collagen, glass, silica, etc.) [62]. First, we measured activated partial thromboplastin time (aPTT—Figure 4A) and prothrombin time (PT—Figure 4B), which reflected the coagulation time for intrinsic and extrinsic pathways, after exposure of platelet-poor plasma (PPP) to GO NPs. PT and aPTT are routine parameters used in clinical practice for blood plasma coagulation (fibrin clot formation) assessment [59]. The normal expected ranges for aPTT and PT are 25.1–36.5 s and 9.4–12.5 s, respectively, as reported previously [15,26]. However, we determined significantly higher values for the control PT (e.g., physiological levels of untreated PPP of six healthy donors) (14.3–20.3) than those reported in the literature, while the control aPTT values (24.3–37.5) were close to the published data. The addition of nGO-PEG NPs with different concentrations to PPP was similar to the pristine nGO effect on aPTT and PT, suggesting that PEGylation had little or no impact on the blood clotting potential of pristine nGO (Figure 4A,B). Treatment with both nGO NPs influenced to a greater degree the intrinsic (aPTT) than the extrinsic (PT) coagulation pathway, probably due to the interaction of GO NPs with the intrinsic coagulation factors. The PT of all samples was very similar and close to that of the controls, suggesting that the exposure to nGO and nGO-PEG had a slight effect on the extrinsic coagulation pathway. In contrast, aPTT values exceeded the normal range and were markedly prolonged in a concentration-dependent way after nGO and nGO NP treatment. Intriguing results were obtained for the samples treated with the highest concentrations of 200 μg/mL NPs. Data analyses proved that the values for PT and aPTT differed significantly among the donors. aPTT values of two of the donors treated with both types of NP were very high compared to the other concentrations, while in four of the donors the coagulation was not measured. PT measurements also demonstrated maximal values in two of the donors, treated with 200 μg/mL nGO and nGO-PEG NPs, and no coagulation in the other four donors. Based on the obtained above results we suggested that NPs result in weaker thrombocyte aggregation and lower fibrinogen levels in plasma.

#### 3.3.2. Effect of nGO and nGO-PEG NPs on Platelet (Thrombocyte) Aggregation

Platelet aggregation together with the adhesion of platelets and release of mediators is one of the main steps of primary haemostasis, resulting in the formation of a soft platelet plug. Therefore, we have studied the aggregation of platelets after exposure to nGO and nGO-PEG NPs using phase-contrast microscopy. Figure 5 shows the phase-contrast micrographs of unstimulated (Figure 5A—control, Figure 5B—nGO-treated and Figure 5C—nGO-PEG) and ADP-stimulated platelets, control (untreated, Figure 5D) and treated with nGO and nGO-PEG NPs with a concentration of 100 μg/mL (Figure 5E,F, respectively). ADP is a physiologic aggregating agent and was used as a positive control. The examination of platelets incubated with nanoparticles revealed the presence of small and large platelet aggregates upon incubation of platelets with NPs. nGO-PEG NPs induced the formation of much larger platelet aggregates than nGO-PEG (Figure 5). Indeed, we found differences in platelet reactivity to both types of nGO NPs. Incubation of pristine nGO NPs with platelets for 15 min in absence of the agonist, ADP, showed that spontaneous platelet aggregation was not detected. The addition of 50 μg/mL ADP at 15 min resulted in the formation of platelet aggregates. PEGylated nGO NPs induced both spontaneous and ADP-stimulated aggregation of human platelets in vitro (Figure 5B,C,E,F). Additionally, nGO-PEG caused aggregation with little or no granular release (Figure 5C,F, see black arrows), while pristine nGO resulted in platelet degranulation and partial aggregation. This is probably because the sharper edges of the pristine nGO mechanically destroy platelets. They lose their aggregation functions, regardless of the higher amount of plasma platelet aggregation factors (ADP, serotonin, fibrinogen, lysosomal enzymes and β-TG) released. In contrast, in nGO-PEG-treated samples the small amount of the released platelet aggregation factors results in the formation of more platelet aggregates. This suggested that blood platelets were easily targeted and activated by GO nanoparticles in vitro [63].

#### 3.3.3. Effect of nGO and nGO-PEG NPs on Fibrinogen Concentration

Since secondary haemostasis is the formation of fibrinogen into fibrin, which evolves the soft platelet plug into a hard, insoluble fibrin clot, we have measured the concentration of fibrinogen (FNG) in the PPP, treated with nGO and nGO-PEG (Figure 4C). Plasma fibrinogen is an essential coagulation protein produced by the liver (MW340 kDa). Fibrin molecules can clump together to form a solid clot, suggesting that an increase in the level of fibrinogen increases the blood viscosity and thus may pose a risk in cases of cardiovascular disease [64]. The typical concentration of FNG in the blood plasma is approximately 2.5 g/L [65]. After GO treatment, the fibrinogen levels in our study were found to decrease significantly in a concentration-dependent manner in comparison to the untreated controls (Figure 4C). Our results are similar to the finding of several other reports demonstrating a positive effect of NPs on FNG levels in the plasma. This suggests that GO NPs could play a positive role in decreasing systemic inflammatory markers, such as fibrinogen, and improving the patient’s systemic health condition [66].

## 4. Conclusions

Our study showed that pristine and PEGylated nGO have no haemolytic activity when used in concentrations up to 50 μg/mL. The higher concentrations of NPs, however, induced dose-dependent haemolysis in human blood as well as alterations in RBCs morphology. Evaluation of plasma coagulation showed that both nGO and nGO-PEG impaired the activated partial thromboplastin time but not prothrombin time of the platelet-poor plasma, suggesting that GO NPs interfere with the intrinsic but not with the extrinsic pathway of blood coagulation. Results from the platelet aggregation indicated that pristine nGO nanoparticles did not induce spontaneously but only ADP-stimulated aggregation of human platelets *in vitro*, while nGO-PEG NPs induced both spontaneously and ADP-stimulated platelet aggregation. Additionally, the exposure to both GO NPs decreased plasma fibrinogen in a concentration-dependent manner, suggesting a positive role of both NPs in decreasing the systemic inflammation. We have further demonstrated a difference between the impact of pristine and PEGylated nGO on blood components. Pristine nGO had higher haemolytic activity and hematotoxicity, indicating that the PEGylation diminished the adverse effects of pristine nGO on human blood, probably by forming a shield around GO NPs. We acknowledge that this hypothesis needs further research to be satisfied.

## Figures and Tables

**Figure 1 materials-14-04853-f001:**
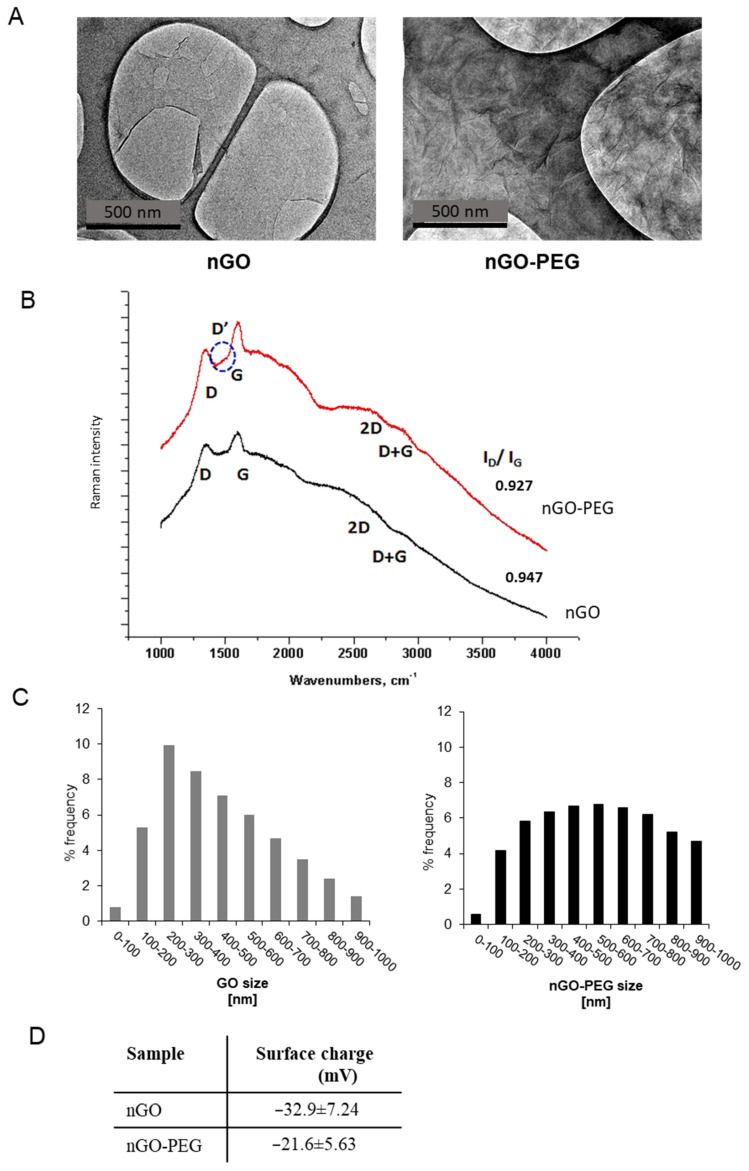
Physiochemical properties of nGO and nGO-PEG NPs: (**A**) TEM analysis of nGO and nGO-PEG after sonication. (**B**) Raman spectra of nGO and nGO-PEG. (**C**) Size distribution of nGO and nGO-PEG after sonication. (**D**) Zeta potential of nGO and nGO-PEG in water solution.

**Figure 2 materials-14-04853-f002:**
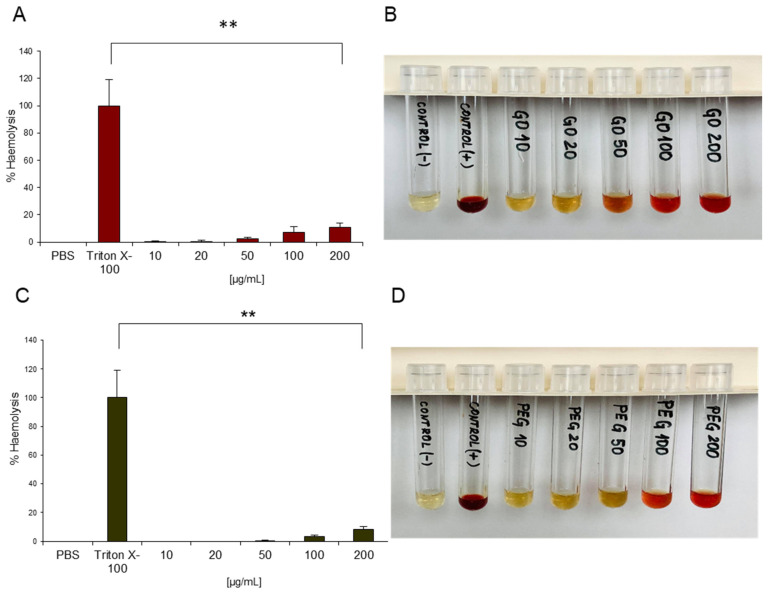
Haemolytic activity of nGO and nGO-PEG NPs. Percentage of haemolysis induced by nGO (**A**) and nGO-PEG (**C**), respectively. Visual inspection via optical images of the tubes containing diluted total blood after exposure to nGO (**B**) or nGO-PEG (**D**) for 3 h after centrifugation. PBS and Triton-X-100 were, respectively, used as negative control and positive control. ** *p* < 0.01 denotes significantly different results from the negative control.

**Figure 3 materials-14-04853-f003:**
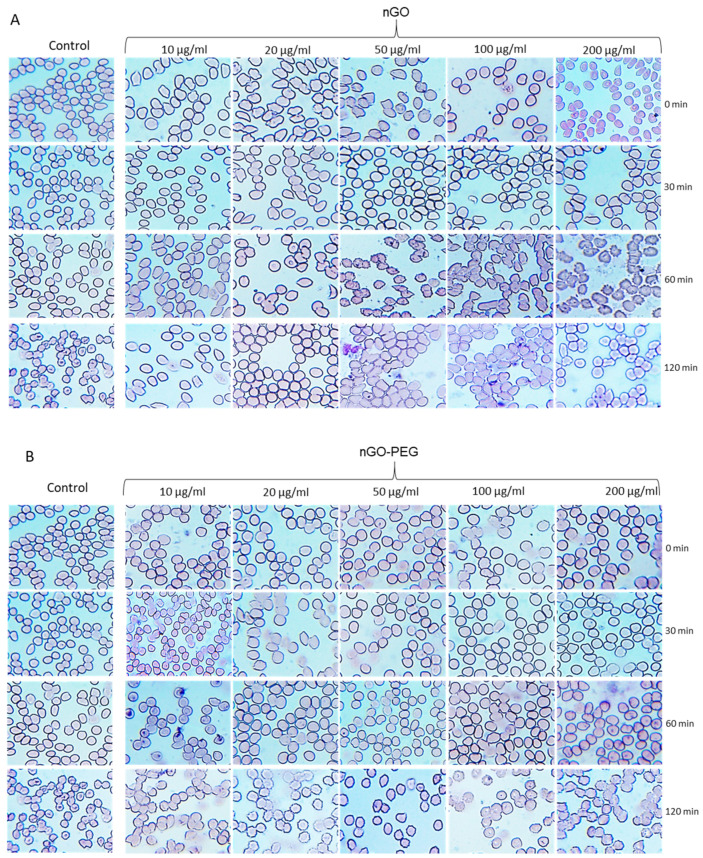
Effects of nGO and nGO-PEG NPs on erythrocyte morphology of RBCs after being treated by different concentrations (0, 10, 20, 50, 100 and 200 μg/mL) of nGO (**A**) and nGO-PEG NPs (**B**) for different periods (0, 30, 60 and 120 min).

**Figure 4 materials-14-04853-f004:**
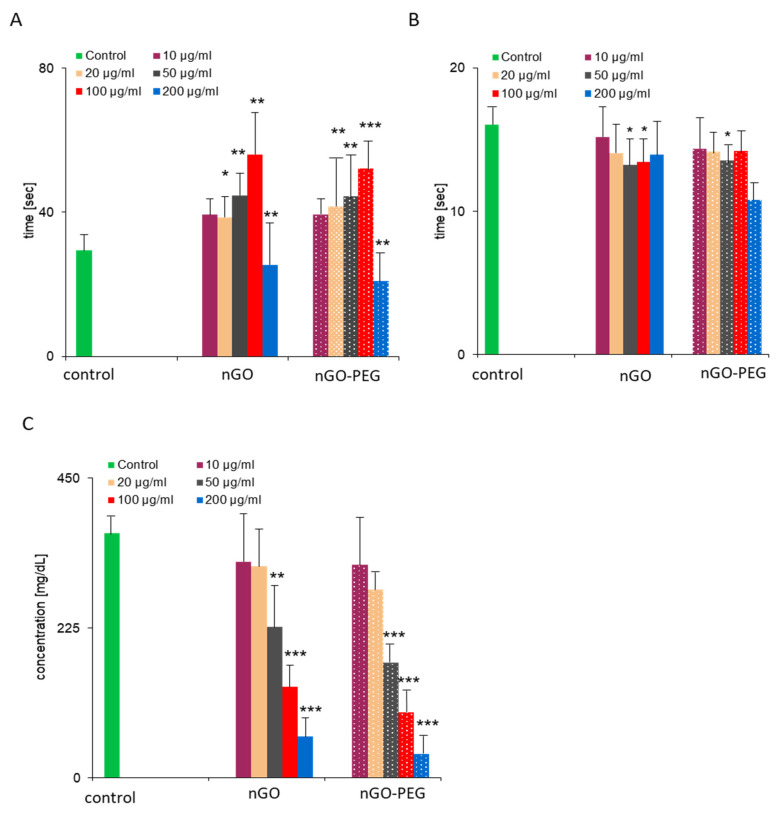
Measurement of plasma coagulation time and fibrinogen concentration: Fresh human plasma was co-incubated with 100 μg/mL of graphene derivatives for 30 min at 37 °C. Plasma coagulation was induced by the addition of sodium citrate and activated partial thromboplastin time (aPTT) (**A**); prothrombin time (PT) (**B**) and fibrinogen concentration (**C**) were recorded, respectively. One-way analysis of variances with Dunnett’s multiple comparison test was applied. Significant results are indicated as * *p* < 0.05; ** *p* < 0.01 and *** *p* < 0.001.

**Figure 5 materials-14-04853-f005:**
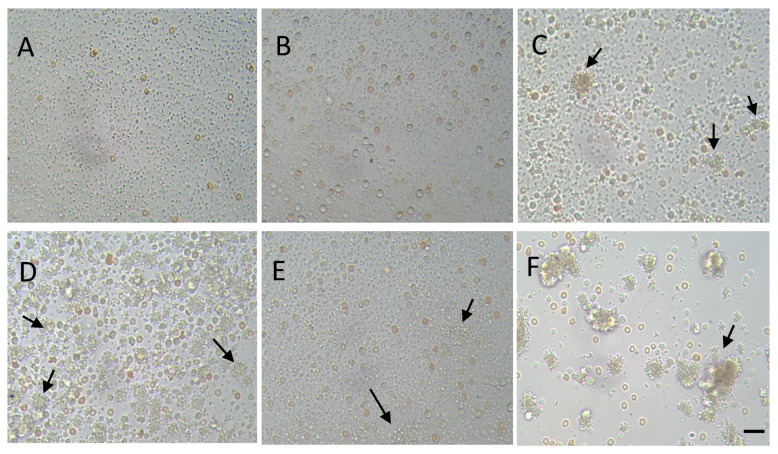
Phase-contrast micrographs of unstimulated (**A**–**C**) and ADP-stimulated (**D**–**F**) platelets, treated with nGO (**B**,**E**) and nGO-PEG (**C**,**F**) NPs. A quantity of 100 μg/mL of nGO and nGO-PEG nanoparticles were co-incubated with platelet-rich plasma and pplatelet activation was induced by addition of 50 µM/mL of adenosine diphosphate (ADP). Arrows show platelet aggregates. Scale bar: 20 μm.

## Data Availability

All data are freely available.

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
