# Peer review of "Impact of Polyethylene Glycol Functionalization of Graphene Oxide on Anticoagulation and Haemolytic Properties of Human Blood"

_materials, 2021, doi:10.3390/ma14174853_

Round 1

Reviewer 1 Report

As the author said that Graphene oxide (GO) is one of the most explored nanomaterials in medical use, it is important to explore and exploit interaction between the particles and the blood components. The research design is appropriate. English must be improved. 

Author Response

Dear reviewer,

Thank you for the prompt and credible revision of our work. We, therefore, believe that the comments by the reviewers have improved the quality of our work.

Herein, in the revised version of our manuscript, we address all comments made by you and all other reviewers.

All suggested edits have been done to the main text.

Regards,

Assoc. prof. Natalia Krasteva

Reviewer:

As the author said that Graphene oxide (GO) is one of the most explored nanomaterials in medical use, it is important to explore and exploit the interaction between the particles and the blood components. The research design is appropriate. English must be improved. 

Answer:

We have proofread the manuscript and have corrected all necessary grammar and stylistic errors. All edits are denoted with track changes. Furthermore, we have edited the Introduction part and made it more concise and comprehensible.

Reviewer 2 Report

PEG and graphene oxide nanoparticles are now commonly used nanomedicine materials. The author of this article used PEG to regulate the anticoagulant and hemolytic effects of graphene oxide. The work of this article is relatively solid, and the experimental data is sufficient and convincing. It is recommended to accept it after a minor revision. 

However, here are some issues should be addressed:

1) PEG and GO were simply mixed. What is the interface relationship between them? Despite of SEM morphology, is there any other evidence that PEG can perfectly package GO?

2) What is the stability of PEG/GO nanoparticles in PBS or other liquids?

3) As your introduction says, cytotoxicity and absorption, especially for nanoparticles, are factors that must be considered for biomaterials. Is there a comparison in the discussion section?

4) As one of the mainstream clinical applications of anticoagulation and hemolysis, wound dressings, i.e. Chen, Zhongda, et al. Materials Science and Engineering: C (2021): 112204., should be briefly introduced in the introduction section.

Author Response

Dear reviewer,

Thank you for the prompt and credible revision of our work. We, therefore, believe that the comments by you have improved the quality of our work.

Herein, in the revised version of our manuscript, we address all comments made by you and all other reviewers.

All suggested edits have been done to the main text and figures.

Regards,

Assoc. prof. Natalia Krasteva

Reviewer:

PEG and graphene oxide nanoparticles are now commonly used Nanomedicine materials. The author of this article used PEG to regulate the anticoagulant and hemolytic effects of graphene oxide. The work of this article is relatively solid, and the experimental data is sufficient and convincing. It is recommended to accept it after a minor revision. 

However, here are some issues should be addressed:

1) PEG and GO were simply mixed. What is the interface relationship between them? Despite of SEM morphology, is there any other evidence that PEG can perfectly package GO?

The question is logical and we have to admit that nowhere in the text we have clearly stated that PEG perfectly packed GO. This has not been proven in either of our experiments. Honestly, we are not sure how exactly PEG packed GO. Therefore, in the current revised manuscript Line 455 has been revised and a sentence stating that we only hypothesize on the way PEG package GO is inserted, please see:

“Pristine nGO had higher haemolytic activity and hematotoxicity indicating that the PEGylation diminished the adverse effects of pristine nGO on human blood probably by forming a shield around GO NPs with the knowledge that this hypothesis needs further research to be satisfied.”

Probably by a computer simulation it could be clarified. In our previous work, we have thoroughly investigated the physicochemical properties of GO and GO-PEG in order to prove the successful PEGylation of GO (https://pubmed.ncbi.nlm.nih.gov/33809878/). We have found an alteration in all measured physicochemical properties of GO after PEGylation pointing to successful PEGylation of GO. Among all used techniques FTIR measurements gave the most adequate answer about what is the interface relationship between GO and PEG. IR spectra demonstrated a reduction of the oxygen-containing groups especially of the carbonyl function groups perhaps because of the establishment of an amide bond (-CO-NH-) in the process of incorporation of PEG-NH2 on the nGO surface edges. This was solid proof that PEG is connected to GO by carboxyl/carbonic groups which are the edges of the nanosheets.

Packaging of GO by PEG is a good prospective analysis in our research and we are grateful for that question. In our future experiments, this will be addressed adequately. For now, the proof that PEG is connected and modifies GO is enough to discuss the results for their biological activity.

2) What is the stability of PEG/GO nanoparticles in PBS or other liquids?

The colloidal stability of nanoparticles in different liquids is a very important issue. Nanoparticles stability is the resultant of the physicochemical properties of the nanomaterials and the acidity of the surrounding media (10.1007/s10853-015-8934-z; 10.1021/acs.langmuir.6b01012; 10.1021/es400483k ). Different salts could disrupt the colloidal stability of graphene materials [10.1021/la404134x], alerting their mobility, and interfering with the dose delivered to the cells [10.1021/nn202355p ]. Many studies stressed the stability of GO and PEG-GO. For example, Liu et al demonstrated that covalently conjugated PEG-GO NPs were stably dispersed in water, phosphate-buffered saline (PBS), complete culture medium and serum whereas, GO settled down partially in PBS and completely in both culture medium [10.1021/ja803688x ].

In our previous publication [10.3390/pharmaceutics13030424] we have evaluated by DLS the stability of nGO and nGO-PEG incubated in the aqueous suspension for ten days under storage conditions (at 4°C) and have found that the zeta potential and the average particle size of nGO-PEG dispersion were stable during incubation time while both measured values of nGO dispersion had changed. Also, we have macroscopically observed nGO and nGO-PEG in DMEM containing 10% FBS under three days of cell exposure and found that nGO aggregated while nGO-PEG were well dispersed. All data showed that PEG functionalization enhanced stability in biological fluids.

3) As your introduction says, cytotoxicity and absorption, especially for nanoparticles, are factors that must be considered for biomaterials. Is there a comparison in the discussion section?

In this work, we have focused on the haemocompatability/haemotoxicity of GO and GO-PEG NPs, and more specifically on their anticoagulation and haemolytic properties. We haven’t done a correlation/comparison analysis between the haemotoxicity of GO-PEG NPs and their cellular uptake because we haven't investigated this and have no results about NPs adsorption/uptake on/in blood cells. We suppose that the increased haemolysis of the blood cells treated with higher NPs concentrations is a result of the higher adsorption of GOs NP on the cell surface or the higher uptake of the NPs in erythrocytes. All these need further studies and we are planning to investigate more in detail the internalization of GO-PEG in different cells, including cancer and blood cells.

4) As one of the mainstream clinical applications of anticoagulation and hemolysis, wound dressings, i.e. Chen, Zhongda, et al. Materials Science and Engineering: C (2021): 112204., should be briefly introduced in the introduction section

We thank you for this suggestion and have included a brief summary of the possible applications of these wound-dressing nanomaterials. Please, see line 69.

Reviewer 3 Report

The manuscript entitled "Impact of polyethylene glycol functionalization of graphene oxide on anticoagulation and haemolytic properties of human
blood" describes a functionalization method for improving graphene properties in such a way that the PEGylation diminished the ad- 433
verse effects of pristine nGO on human blood by forming a shield around GO NPs. The manuscript has scientific soundness and it is addressed mostly to biologists and medical researchers. The findings reported by the authors has great importance for the bio-medical applications of graphene.

I have only a few recommendations:

Recommendations:

  1. Can you please choose other colors for 20 and 50 ug/mL to section 3.3. Anticoagulation properties of nGO and nGO-PEG NPs? The colors you used are not so visible in the graphs and it is quite hard to follow.
  2. Why some of the images from figure 3 A and B are blurred?
  3. So, from TEM characterization and size distribution, one can observe that by PEGylation your graphene oxide NPs resulted in greater sizes but with a more uniform size distribution. Can you explain how these properties influenced the results in such a way that graphene is less hemotoxic when functionalized with PEG? I consider that some correlations between the materials' properties and the hemolytic results should be made and introduced in the manuscript.
  4. In Figure 1 there is a switch between the letters of the images, there is no correspondence between the letters for each image and the figure caption text. Please correct this!
  5. Figure 1, Raman spectra (B is the correct letter), maybe you could increase the part of the Raman spectra for the GO-PEG NPs where you observe the D' band? It would be more visible for the readers because in this form it is hard to observe anything.

Author Response

Dear reviewer,

Thank you for the prompt and credible revision of our work. We, therefore, believe that the comments by you have improved the quality of our work.

Herein, in the revised version of our manuscript, we address all comments made by you following their logic. All suggested edits have been done to the main text and figures and all comments and concerns are addressed.

Looking forward to your positive evaluation and for the place of our work in Materials.

Regards,

Assoc. prof. Natalia Krasteva

Reviewer:

The manuscript entitled "Impact of polyethylene glycol functionalization of graphene oxide on anticoagulation and haemolytic properties of human blood" describes a functionalization method for improving graphene properties in such a way that the PEGylation diminished the adverse effects of pristine nGO on human blood by forming a shield around GO NPs. The manuscript has scientific soundness and it is addressed mostly to biologists and medical researchers. The findings reported by the authors has great importance for the bio-medical applications of graphene.

I have only a few recommendations:

Recommendations:

  1. Can you please choose other colors for 20 and 50 ug/mL to section 3.3. Anticoagulation properties of nGO and nGO-PEG NPs? The colors you used are not so visible in the graphs and it is quite hard to follow.

Please correct this!

Answer:

Yes, we have changed the colours and edited the figure.

2. Why some of the images from figure 3 A and B are blurred?

Answer:

All images are edited and sharpened. The blurred ones are replaced with images of better quality.

3. So, from TEM characterization and size distribution, one can observe that by PEGylation your graphene oxide NPs resulted in greater sizes but with a more uniform size distribution. Can you explain how these properties influenced the results in such a way that graphene is less hemotoxic when functionalized with PEG? I consider that some correlations between the materials' properties and the hemolytic results should be made and introduced in the manuscript.

Answer:

This comment allowed us to rethink our results and formulate a better explanation of the observed results. We, therefore, edited the text in Line 302:

Please see:

“The differences in haemolytic properties of nGO and nGO-PEG may be referred to as the differences in their size and particle size distribution. Particle size together with chemical composition is one of the most important characteristics which influence interactions with cells [37, 38]. The observed lower haemolytic activity of nGO-PEG as compared to those of nGO is due to the larger particle size and distribution resulting in a small number of NPs contacting with cells and destroying erythrocyte membrane integrity. Opposite, in the case of nGO NPs, a great number of NPs have a very small size. These NPs can damage very easily cell membrane because of the sharper edges of pristine nGO NPs which results in a higher degree of haemolysis.”

4. Figure 1, Raman spectra (B is the correct letter), maybe you could increase the part of the Raman spectra for the GO-PEG NPs where you observe the D' band? It would be more visible for the readers because in this form it is hard to observe anything.

Answer:

Thank you for this remark!

We have edited the figure accordingly.